# World-Env: Leveraging World Model as a Virtual Environment for VLA Post-Training

## Abstract

Vision-Language-Action (VLA) models trained via imitation learning suffer from significant performance degradation in data-scarce scenarios due to their reliance on large-scale demonstration datasets. Although reinforcement learning (RL)-based post-training has proven effective in addressing data scarcity, its application to VLA models is hindered by the non-resettable nature of real-world environments. This limitation is particularly critical in high-risk domains such as industrial automation, where interactions often induce state changes that are costly or infeasible to revert. Furthermore, existing VLA approaches lack a reliable mechanism for detecting task completion, leading to redundant actions that reduce overall task success rates. To address these challenges, we propose World-Env, an RL-based post-training framework that replaces physical interaction with a low-cost, world model-based virtual simulator. World-Env consists of two key components: (1) a video-based world simulator that generates temporally consistent future visual observations, and (2) a vision-language model (VLM)-guided instant reflector that provides continuous reward signals and predicts action termination. This simulated environment enables VLA models to safely explore and generalize beyond their initial imitation learning distribution. Our method achieves notable performance gains with as few as five expert demonstrations per task. Experiments on complex robotic manipulation tasks demonstrate that World-Env effectively overcomes the data inefficiency, safety constraints, and inefficient execution of conventional VLA models that rely on real-world interaction, offering a practical and scalable solution for post-training in resource-constrained settings.

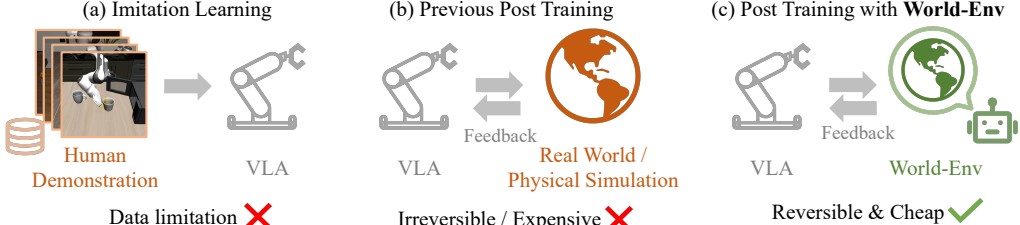

Figure 1: Comparison of three VLA training paradigms: (a) Imitation learning suffers from poor generalization under data scarcity. (b) Prior RL-based post-training methods require real-world interaction, which is often infeasible due to non-resettable state transitions (e.g., object drop or collision). (c) Our proposed World-Env enables post-training via simulated rollouts using a world model, eliminating the need for physical interaction and supporting safe, efficient exploration even with minimal expert demonstrations.

## 1 Introduction

Vision-Language-Action (VLA) models have emerged as a central paradigm for autonomous agents, enabling end-to-end mapping from high-level language instructions to low-level motor commands by integrating vision, language, and control. These models have demonstrated considerable promise in robotic manipulation (Kim et al., 2024; Black et al., 2024), autonomous driving (Yurtsever et al.,

2020), and navigation (Hong et al., 2021). Most existing approaches rely on supervised fine-tuning through imitation learning, building upon pre-trained vision-language models (Touvron et al., 2023a) to align semantic intent with physical execution via cross-modal representations.

However, imitation learning methods (Kim et al., 2025) are inherently constrained by the limited availability of high-quality demonstrations. In many real-world scenarios, collecting diverse and safe human demonstrations is prohibitively expensive and often infeasible due to safety concerns and environmental complexity. Furthermore, such methods generalize poorly to novel tasks or unseen objects, and their performance degrades under few-shot conditions.

To overcome these shortcomings, recent works (Tan et al., 2025; Lu et al., 2025) have turned to reinforcement learning (RL) (Rafailov et al., 2023) to enable agents to learn through interaction. Current RL strategies fall into two categories. The first involves real-world learning with human feedback, which captures realistic environmental dynamics but suffers from non-resettable interactions, high trial costs, and limited reproducibility, rendering it unsuitable for safety-critical applications. The second relies on simulator-based learning, which avoids physical risks but introduces other challenges, including substantial development effort, limited sim-to-real transfer, and difficulty adapting to new objects or dynamic scene changes, thereby restricting its practical applicability.

These limitations motivate us to think about a question: *Is there an "ideal testbed" that avoids real-world risks while providing greater flexibility and richer semantic understanding than conventional simulators?* We find that video-based world model offers a promising solution. Equipped with action-conditioned future prediction and a persistent latent scene representation, world model can generate visually plausible future image sequences, allowing safe, low-cost simulation of action outcomes, as well as policy exploration and refinement without physical interaction.

In this work, we introduce World-Env, a world model-based reinforcement learning framework that improves policy generalization under data scarcity while respecting real-world safety constraints, as shown in Figure 1. World-Env consists of two components. The first is a video-based world simulator that functions as an interactive future-frame predictor, synthesizing action-conditioned image sequences that capture post-interaction object states and surrounding scene structure. The second is a VLM-guided instant reflector that functions as a semantics-aware reward module. It provides continuous reward signals by evaluating the semantic alignment between predicted visual frames and the input language instruction. This assessment supports policy optimization and enables real-time detection of task completion. Upon confirming successful execution (e.g., when the goal state is reached), the reflector immediately terminates the action sequence to prevent redundant or disruptive subsequent actions. The framework delivers three principal benefits: (1) efficient generalization from minimal expert demonstrations, (2) safe controllability through risk-free virtual exploration, and (3) language-aligned termination via VLM-driven reasoning.

In summary, our contributions are:

- We propose World-Env, a world model-based framework that enables low-cost, safe reinforcement learning post-training for VLA policies under extreme data scarcity, eliminating the need for real-world interaction.

- World-Env integrates a video-based world simulator and a VLM-guided instant reflector to jointly provide temporally consistent visual observations and continuous reward signals, forming a self-contained virtual environment that supports effective policy exploration.

- We introduce a real-time termination mechanism via the instant reflector, which dynamically assesses task completion by evaluating semantic alignment between predicted visual trajectories and language instructions, thereby preventing redundant post-success actions.

## 2 RELATED WORK

**Vision-Language-Action Models.** Leveraging advancements in pre-trained vision foundation models (Radford et al., 2021; Oquab et al., 2023; Dosovitskiy et al., 2020), large language models (LLMs) (Brown et al., 2020; Touvron et al., 2023b; Bai et al., 2023), and vision-language models (VLMs) (Alayrac et al., 2022; Li et al., 2022; 2023; Dai et al., 2023; Liu et al., 2023c;b), Vision-Language-Action (VLA) frameworks (Chi et al., 2023; Team et al., 2024; Kim et al., 2024; Bu et al.,

2025) have emerged as a powerful approach for embodied intelligence. These models bridge the gap between high-dimensional sensory inputs and physical-world actions by processing multimodal signals, such as camera feeds, sensor data, and natural language instructions, and translating them into actionable outputs for robotic systems. DiffusionPolicy (Chi et al., 2023) proposes a diffusion-based policy that generates robot actions through a conditional denoising diffusion process in the action space, iteratively refining actions based on visual observations. OpenVLA (Kim et al., 2024) integrates robotic actions into a language modeling framework by mapping action sequences to discrete tokens within a large language model. The recent work OpenVLA-OFT (Kim et al., 2025) further converts discrete action sequences into continuous representations, achieving improved inference efficiency and task performance.

**Reinforcement Learning for VLA Systems.** Recent advances in reinforcement learning (RL) (Schulman et al., 2017; Rafailov et al., 2023) have demonstrated considerable potential in enhancing decision-making capabilities of large language models (LLMs) (Guo et al., 2025; Lightman et al., 2023; Ouyang et al., 2022; Lee et al., 2023). This progress has spurred growing interest in applying RL to Vision-Language-Action (VLA) systems (Tan et al., 2025; Lu et al., 2025; Chandra et al., 2025; Jiang et al., 2025a), where adaptive behavior is essential. Unlike supervised fine-tuning (SFT), which replicates static demonstrations, RL enables agents to refine policies through interaction, optimizing actions to maximize task-oriented rewards. This paradigm supports autonomous exploration, reward-driven adaptation, and improved robustness to partial observability, allowing VLA models to generalize to unseen scenarios while reducing reliance on costly human demonstrations. However, existing RL-based VLA methods typically require real-world interaction, which is often infeasible in high-risk or non-resettable scenarios.

**World Models.** World models (Assran et al., 2025; Ball et al., 2025), which are learned dynamical simulators that approximate environmental dynamics, have become foundational for safe and sample-efficient agent training. Early works (Hafner et al., 2019; 2020; 2025) demonstrated the effectiveness of model-based reinforcement learning in virtual environments, enabling agents to plan via imagined trajectories without real-world interaction. Recent advances, such as TD-MPC2 (Hansen et al., 2024), have improved scalability and policy learning efficiency across multi-task and multi-domain settings. Similarly, PWM (Georgiev et al., 2025) leverages pre-trained world models and first-order optimization to handle high-dimensional action spaces in multi-task RL. However, these methods typically rely on on-policy data, limiting their generalization to specific environments and downstream tasks. Building on advances in diffusion-based video generation (Ho et al., 2020; Rombach et al., 2022; Blattmann et al., 2023; Yang et al., 2024; Wan et al., 2025; Xing et al., 2024), we propose a framework that trains a world model on offline demonstration data and keeps it fixed during policy learning to predict future visual observations for VLA models. This decouples world model training from policy exploration, enabling broader applicability in resource-constrained or high-risk scenarios.

## 3 PRELIMINARY

**Vision-Language-Action Models.** Vision-language-action (VLA) models bridge natural language instructions with robotic control by translating semantic goals into low-level actions while grounding language in multimodal observations. Following recent VLA frameworks such as OpenVLA-OFT (Kim et al., 2025), the policy is implemented as a deterministic mapping that leverages a pretrained vision-language model to extract multimodal features, followed by a lightweight action head for continuous control. Specifically, given a history of RGB observations $\mathbf{o}_{1:t}$, proprioceptive states $\mathbf{s}_{1:t}$ (e.g., joint angles or end-effector poses), and a language instruction $\mathbf{g}$, the policy predicts a deterministic action $\mathbf{a}_t \in \mathbb{R}^D$ as:

$$\mathbf{a}_t = \pi_\theta(\mathbf{o}_{1:t}, \mathbf{s}_{1:t}, \mathbf{g}), \tag{1}$$

where $\pi_\theta$ denotes a deterministic policy parameterized by a finetuned foundation model and a trainable action head.

**Reinforcement Learning.** Reinforcement learning (RL) formulates decision-making as a Markov Decision Process (MDP): $\mathcal{M} = (\mathcal{S}, \mathcal{A}, \mathcal{P}, \mathcal{R}, \gamma)$, where $\mathcal{S}$ is the state space (comprising visual observations $\mathbf{o}_t$ and proprioceptive states $\mathbf{s}_t$), $\mathcal{A}$ is the action space (e.g., continuous control commands

Figure 2: **Overview of World-Env.** Our framework comprises: (1) a *Training Data Strategy* that augments human demonstrations trajectories with VLA self-explored trajectories to train the World-Env Simulator; (2) an *Optimization Loop* where the VLA model generates actions, the simulator predicts future observations, and the World-Env Reflector generates feedback; and (3) *Reward & End Signal* provides trajectory-wise reward and end signals for RL optimization.

$\mathbf{a}_t \in \mathbb{R}^D$), $\mathcal{P}$ denotes the transition dynamics, $\mathcal{R}$ is the reward function, and $\gamma \in [0, 1]$ is the discount factor. The objective is to learn a policy $\pi_\theta(\mathbf{o}_{1:t}, \mathbf{s}_{1:t}, \mathbf{g})$ that maximizes the expected return:

$$J(\pi_\theta) = \mathbb{E}_{\pi_\theta} \left[ \sum_{t=0}^{T} \gamma^t r_t \right],$$

where $r_t = \mathcal{R}(\mathbf{o}_{1:t}, \mathbf{g})$. In practice, policy gradient methods often introduce stochasticity during training to enable exploration. The policy is updated using gradients of the form:

$$\nabla_\theta J(\pi_\theta) = \mathbb{E}_{\mathbf{a}_t \sim \pi_\theta} \left[ \nabla_\theta \pi_\theta(\mathbf{o}_{1:t}, \mathbf{s}_{1:t}, \mathbf{g}) \cdot A(\mathbf{o}_{1:t}, \mathbf{a}_{1:t}) \right], \tag{2}$$

where $A(\cdot)$ is the advantage function that evaluates action quality relative to a baseline.

## 4 METHOD

Figure 2 presents the overview of our framework. Prior VLA approaches (Kim et al., 2024; 2025) typically rely on either real-world interaction or conventional simulators to provide observations for action prediction. In contrast, our framework eliminates the need for physical interaction by leveraging a **video-based world simulator** that generates temporally consistent future visual observations at low cost. Specifically, the deterministic VLA policy $\pi_\theta$ maps the current RGB observation $\mathbf{o}_t$, language instruction $\mathbf{g}$, and proprioceptive state $\mathbf{s}_t$ (comprising the 6D end-effector pose and 1D gripper state) to a continuous action $\mathbf{a}_t$. The next proprioceptive state $\mathbf{s}_{t+1}$ is then computed deterministically from $\mathbf{s}_t$ and $\mathbf{a}_t$ using forward kinematics. The world simulator takes the executed action $\mathbf{a}_t$ and the resulting proprioceptive state $\mathbf{s}_{t+1}$ as inputs and predicts the subsequent visual observation $\mathbf{o}_{t+1}$. This imagined observation, together with $\mathbf{s}_{t+1}$, is fed back into the VLA policy to predict the next action $\mathbf{a}_{t+1}$. The rollout terminates either when the maximum timestep is reached or when the **VLM-guided instant reflector**, which evaluates semantic alignment between the predicted visual trajectory and the language instruction, confirms task success and issues a termination signal. During training, we collect $N$ simulated trajectories from this virtual environment and use them for reinforcement learning (RL) optimization of the VLA policy within World-Env.

### 4.1 VIDEO-BASED WORLD SIMULATOR

Our world simulator is built upon the EVAC framework (Jiang et al., 2025b). During rollout, the simulator takes the executed action $\mathbf{a}_t$ and the resulting proprioceptive state $\mathbf{s}_{t+1}$ as inputs to predict the next visual observation $\mathbf{o}_{t+1}$. The proprioceptive state $\mathbf{s}_{t+1}$ comprises a 3D position vector $\mathbf{x}_{t+1} \in \mathbb{R}^3$, a 3D rotation vector $\mathbf{q}_{t+1} \in \mathbb{R}^3$ (represented in axis-angle format), and a 1D gripper state $p_{t+1} \in [0, 1]$.

Following Jiang et al. (2025b), we render an action map by projecting the proprioceptive state $\mathbf{s}_{t+1}$ onto the image plane. This action map consists of a foreground marker indicating the projected

pose and a black background to enhance visual contrast. The action map, together with the history observation, is injected into the EVAC world model as pixel-level conditioning. The EVAC model then generates the future observation $\mathbf{o}_{t+1}$ using a diffusion-based image generation module. For further architectural details, we refer readers to Jiang et al. (2025b).

To train the world model, we find that relying solely on expert demonstrations from the LIBERO benchmark (Liu et al., 2023a) limits generalization to unseen state-action sequences. To address this, we augment the training data by enabling autonomous exploration in the LIBERO simulator. Specifically, we deploy the supervised fine-tuned OpenVLA-OFT policy (Kim et al., 2025) to predict actions and execute them in the simulator, which yields the corresponding next proprioceptive state $\mathbf{s}_{t+1}$ and observation $\mathbf{o}_{t+1}$. To further enhance data diversity, we introduce controlled stochasticity by training a scale head that predicts the log-scale parameter $\boldsymbol{\beta}_t$ of a Laplace distribution, with the OpenVLA-OFT action $\boldsymbol{\mu}_t$ as the location parameter: $\mathbf{a}_t \sim \text{Laplace}(\boldsymbol{\mu}_t, \boldsymbol{\beta}_t)$. These perturbed actions are executed to collect additional $(\mathbf{o}_t, \mathbf{s}_t, \mathbf{a}_t, \mathbf{s}_{t+1}, \mathbf{o}_{t+1})$ transition tuples. Finally, we combine these autonomously collected trajectories with the original human-demonstrated successful trajectories from LIBERO (Liu et al., 2023a) to form a diverse and robust training dataset for the world simulator. Additional analysis of the world simulator and network architecture are provided in the supplementary material.

## 4.2 VLM-GUIDED INSTANT REFLECTOR

Previous methods (Tan et al., 2025; Lu et al., 2025) rely on simulators to provide binary task success signals, using sparse discrete rewards for RL post-training. These approaches suffer from a key limitation: the lack of termination-aware feedback, causing policies to often continue executing redundant actions after task completion (e.g., over-scooping after object placement). To address this, we propose a VLM-guided instant reflector that leverages LLaVA (Liu et al., 2023c), a pretrained vision-language model, to provide a continuous-valued reward signal.

Given a video of imagined observations $\mathbf{o}_{1:t}$ and a language instruction $\mathbf{g}$, the instant reflector predicts a step-wise reward $R(\mathbf{o}_{1:t}, \mathbf{g}) \in [0, 1]$ for each time step $t$, which estimates the probability that the task has been successfully completed by time $t$. The architecture consists of a frozen vision encoder $\mathcal{E}_{\text{vision}}$ that extracts patch embeddings from video frames, a frozen LLM $\mathcal{E}_{\text{LLM}}$ that performs cross-modal reasoning over the visual-language sequence, and a lightweight reward head $\mathcal{R}_\theta$ that computes:

$$R(\mathbf{o}_{1:t}, \mathbf{g}) = \sigma(\mathcal{R}_\theta(h_t)), \tag{3}$$

where $\sigma$ is the sigmoid function and $h_t$ is the pooled multimodal embedding from the LLM at time $t$. The termination signal is triggered at the timestep $t$ where $R(\mathbf{o}_{1:t}, \mathbf{g}) > \eta$, with threshold $\eta = 0.5$.

For training, we utilize per-frame binary success labels: for each trajectory, every timestep $t$ is annotated with $y_t \in \{0, 1\}$, indicating whether the task is completed at or before $t$. These labels are derived from two sources: (1) expert trajectories from the LIBERO dataset (Liu et al., 2023a), where success is determined by task-specific criteria, and (2) policy-generated trajectories collected in simulator (Section 4.1), labeled using an oracle that monitors ground-truth task states in the simulator. The reward head $\mathcal{R}_\theta$ is trained with binary cross-entropy (BCE) loss:

$$\mathcal{L} = \text{BCE}\big(R(\mathbf{o}_{1:t}, \mathbf{g}), y_t\big).$$

This supervision enables the reflector to recognize task completion as soon as it occurs, rather than relying on trajectory-level signals. During RL, we use the reward sparsely: the return is computed using a single reward assigned at the termination timestep (or at $T$ if no termination occurs), ensuring compatibility with standard policy gradient methods. This design allows World-Env to simultaneously support real-time termination and efficient policy learning, effectively addressing the execution inefficiency of prior VLA post-training approaches (Tan et al., 2025; Lu et al., 2025). More details can be found in the supplementary material.

## 4.3 POST TRAINING OF VLA MODEL

Our reinforcement learning pipeline employs a PPO-style objective with continuous reward signals; the full algorithm is provided in the supplementary material. Following Tan et al. (2025), training proceeds in three stages: rollout generation, advantage estimation, and policy optimization.

Table 1: **Success rate comparison on the LIBERO benchmark.** We report success rates for each method using the same setting with only 5 demonstrations per task.

| Method | LIBERO-Goal | LIBERO-Object | LIBERO-Spatial | LIBERO-Long | Average |
|---|---|---|---|---|---|
| $\pi_0$ (Black et al., 2024) | 67.6 | 68.4 | 80.2 | 28.2 | 61.1 |
| $\pi_0$+FAST (Pertsch et al., 2025) | 59.2 | 76.8 | 59.2 | 24.8 | 55.0 |
| OpenVLA (Kim et al., 2024) | 73.2 | 55.0 | 82.4 | 32.2 | 60.7 |
| UniVLA (Bu et al., 2025) | 82.0 | 76.2 | 84.4 | 56.4 | 74.75 |
| OpenVLA-OFT (Kim et al., 2025) | 84.0 | 74.2 | 84.2 | 57.0 | 74.85 |
| OpenVLA-OFT + Post-training (Ours) | **86.4** | **86.6** | **87.6** | **57.8** | **79.6** |

During rollout, we generate trajectories $\tau = (\mathbf{o}_{1:T}, \mathbf{s}_{1:T}, \mathbf{g}, \mathbf{a}_{1:T})$ using the world simulator (Section 4.1). Starting from an initial observation $\mathbf{o}_1$, proprioceptive state $\mathbf{s}_1$, and language instruction $\mathbf{g}$, the deterministic VLA policy $\pi_\theta$ predicts a base action $\boldsymbol{\mu}_t = \pi_\theta(\mathbf{o}_{1:t}, \mathbf{s}_{1:t}, \mathbf{g})$. A separate scale head, trained to model action uncertainty, outputs a log-scale parameter $\boldsymbol{\beta}_t$. Together, they define a factorized Laplace distribution, from which the executed action $\mathbf{a}_t$ is sampled. This enables adaptive, uncertainty-aware exploration. The world simulator then predicts the next observation $\mathbf{o}_{t+1}$ using proprioceptive state $\mathbf{s}_{t+1}$. The VLM-guided instant reflector evaluates the partial visual trajectory $\mathbf{o}_{1:t+1}$ and outputs a step-wise reward $R(\mathbf{o}_{1:t+1}, \mathbf{g}) \in [0, 1]$. Rollout terminates either at the maximum timestep $T$ or when $R(\mathbf{o}_{1:t+1}, \mathbf{g}) > \eta$. For RL, we assign a single trajectory-wise reward $R_n = R(\mathbf{o}_{1:t_{\text{end}}}, \mathbf{g})$, where $t_{\text{end}}$ is the termination or final timestep.

We adopt Leave-One-Out Proximal Policy Optimization (LOOP) (Chen et al., 2025) that combines RLOO (Ahmadian et al., 2024) based advantage estimation and PPO (Schulman et al., 2017) for policy updating. For each initial state, we generate $N$ rollouts $\{\tau_1, \ldots, \tau_N\}$ using a fixed behavior policy $\pi_\phi$ (the policy at the beginning iteration). Each trajectory receives a scalar reward $R_n$ from the instant reflector. The RLOO baseline for trajectory $n$ is the average reward of the other $N-1$ rollouts:

$$b_n = \frac{1}{N-1} \sum_{j \neq n} R_j, \quad A_n = R_n - b_n, \tag{4}$$

where $A_n$ is the trajectory-wise advantage.

To update the policy, we treat both the current and behavior policies as inducing stochastic action distributions via their action and scale heads. The importance ratio at timestep $t$ of trajectory $n$ is computed as:

$$r_{t,n} = \frac{p_\theta(\mathbf{a}_{t,n} \mid \mathbf{o}_{t,n}, \mathbf{s}_{t,n}, \mathbf{g}_n)}{p_\phi(\mathbf{a}_{t,n} \mid \mathbf{o}_{t,n}, \mathbf{s}_{t,n}, \mathbf{g}_n)},$$

where $p_\theta$ and $p_\phi$ denote the action distributions induced by the current policy $\pi_\theta$ and behavior policy $\pi_\phi$, respectively, each modeled as a product of independent Laplace distributions over action dimensions. The policy is optimized via the clipped PPO objective:

$$\mathcal{L}_{\text{PPO}} = -\frac{1}{\sum_n T_n} \sum_{n=1}^{N} \sum_{t=1}^{T_n} \min\left(r_{t,n} A_n, \text{clip}(r_{t,n}, 1-\epsilon, 1+\epsilon) A_n\right), \tag{5}$$

with $T_n$ denotes the length of trajectory $n$ and $\epsilon$ refers to the clipping threshold. Note that the advantage $A_n$ is broadcasted to all timesteps within trajectory.

Unlike prior methods that use binary rewards ($R \in \{0, 1\}$) and require balanced success/failure rollouts for stable training, our continuous reward signal ($R \in [0, 1]$) provides finer-grained feedback. This helps enhance rollout efficiency and training stability, particularly in data-scarce settings.

## 4.4 IMPLEMENTATION DETAILS

All our experiments are conducted on 8 NVIDIA H20 GPUs (96 GB memory each). We adopt LoRA (Hu et al., 2022) with rank 32 for parameter-efficient fine-tuning of the vision-language backbone, while the action head and scale head are trained with full parameters. We use a batch size of 4. The LoRA adapters are optimized with a learning rate of $1 \times 10^{-4}$, and the action/scale heads are trained with a learning rate of $1 \times 10^{-5}$. We set the number of rollouts per iteration to $N = 8$ and the PPO clipping threshold to $\epsilon = 0.1$.

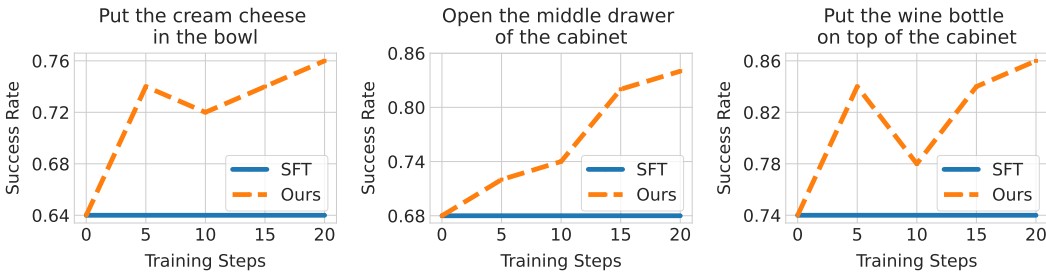

Figure 3: **Comparison between our method and SFT on multi-goal tasks.** Note, all results are collected every 5 training steps for three distinct goals.

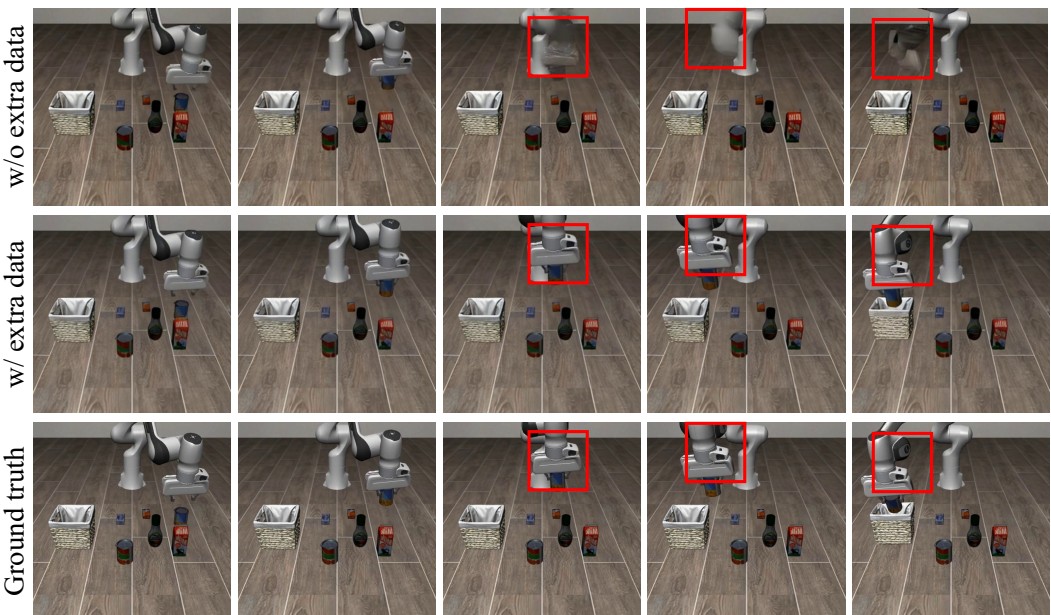

Figure 4: **Rendering comparison of world simulator trained with and without extra data.**

## 5 EXPERIMENTS

Table 2: **Ablation studies.** We evaluate how the extra training data for world simulator learning and the reward head for trajectory scoring affect the performance of our method.

| Extra Data | Reward Head | LIBERO Goal | LIBERO Object | LIBERO Spatial | LIBERO Long |
|:---:|:---:|:---:|:---:|:---:|:---:|
| | | 68.4 | 75.2 | 73.2 | 42.2 |
| ✓ | | 79.8 | 81.8 | 78.4 | 44.6 |
| | ✓ | 68.8 | 76.4 | 74.4 | 43.8 |
| ✓ | ✓ | **86.4** | **86.6** | **87.6** | **57.8** |

**Benchmark.** We evaluate our model on the LIBERO benchmark (Liu et al., 2023a), a simulation-based robotic learning platform designed for vision-language manipulation tasks. The benchmark includes four task suites targeting distinct cognitive challenges: LIBERO-Spatial focusing on spatial reasoning via object arrangement; LIBERO-Goal assessing goal-conditioned planning with end-state requirements; LIBERO-Object testing object-centric manipulation across categories; LIBERO-10 (LIBERO-Long) addressing prolonged sequential decision-making. Each suite contains 10 tasks with 50 trajectories for training and 50 for testing per task; we train OpenVLA-OFT using only 5 trajectories from the training split to validate performance under extreme data scarcity, while evaluating on the full trajectory test split to demonstrate the generalization capability.

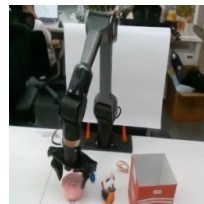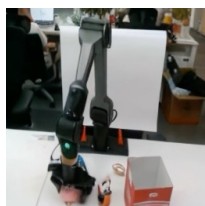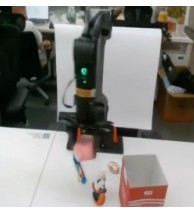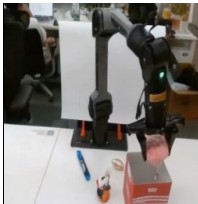

Figure 5: **Real-world rendering results of world simulator.** We show a video sequence generated by our world simulator in real-world scene.

**Baselines.** We compare our method with five state-of-the-art VLA frameworks including $\pi_0$ (Black et al., 2024), $\pi_0$ + FAST (Pertsch et al., 2025), OpenVLA (Kim et al., 2024), UniVLA (Bu et al., 2025), and OpenVLA-OFT (Kim et al., 2025). All methods are trained with standard supervised fine-tuning (SFT). For fair evaluation, all baselines are retrained under identical 5-trajectory per-task constraints, with performance metrics reported on the complete test set.

## 5.1 COMPARISON WITH STATE-OF-THE-ART METHODS

Table 1 presents success rate comparison between our method and the baseline models. As shown, our method gains higher task success rate, demonstrating the effectiveness of our proposed post-training strategy. Figure 3 further compares our method and the supervised fine-tuning (SFT) baseline on multi-goal tasks, where we can see that our approach achieves superior performance within only 20 training steps, clearly outperforming the compared SFT model. This rapid convergence and early dominance highlight the efficiency and effectiveness of our method in learning conditioned policies with minimal training iterations.

## 5.2 ABLATION STUDIES

**Effect of World Simulator.** We investigate how the generative capabilities of world simulator affect the performance of our method. Figure 4 evaluates two world simulator variants: (1) w/o extra: trained solely on human-annotated successful trajectories, (2) w/ extra: enhanced with our collected data containing both successful and failed trajectories. As shown, the model trained without extra data struggles with object tracking, particularly when the VLA model's action predictions deviate from ideal trajectories. This is because the world simulator only observes successful interactions during training, making it unable to simulate complex object states caused by suboptimal actions. In contrast, our model demonstrates significant improvements in robotic arm tracking precision and object interaction fidelity. Table 2 quantitatively validates these observations: when the world simulator generates low-quality images, VLA training effectiveness drops. This correlation highlights the importance of diverse training data in building robust world simulator that can handle real-world action variations. Figure 5 further manifests the simulator's ability to generate photorealistic observations with accurate physical interactions, showing that our method can be adopted in real-world scene application. Please see also the supplementary material for video results.

**Effect of Instant Reflector.** We investigate the effect of instant reflector on our framework's performance. As summarized in Table 2, we perform evaluation for two strategies: (1) w/o reward head: Direct use of pre-trained VLMs with prompt-based binary classification (yes/no) (2) w/ reward head: We integrate a trainable reward head that scores action sequences on a continuous scale. The difference lies in how each approach assesses alignment between generated video sequences and text instructions. While the baseline leverages VLMs' inherent language understanding capabilities through fixed prompts, our method explicitly trains the reward head to quantify task completion progress via scalar scores. Experimental validation demonstrates that a naive use of pre-trained VLMs brings weak performance gains and may degrade VLA model learning in complex scenarios. This limitation stems from the mismatch between VLMs' general language vision alignment and the specific action evaluation requirements. In contrast, our reward head, trained on diverse success/failure trajectories, achieves higher accuracy in distinguishing successful actions.

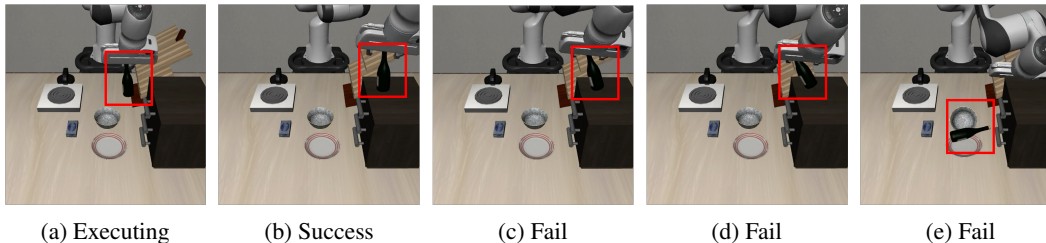

| (a) Executing | (b) Success | (c) Fail | (d) Fail | (e) Fail |

Figure 6: **Post-success failure in VLA execution.** An illustrative example for "put the wine bottle on top of the cabinet" shows the VLA model completes the task (frames a-b), but fails due to delayed termination (frames c-e), validating the necessity of dynamic termination mechanism.

Table 3: **Comparison of task termination strategy under realistic feedback constraints.** Note, all compared methods are evaluated under the setting where ground-truth termination feedback is unavailable, while our method autonomously detects task completion via the proposed reward model. Success rates are measured when reaching the maximum action steps.

| Method | LIBERO-Goal | LIBERO-Object | LIBERO-Spatial | LIBERO-Long | Average |
|---|---|---|---|---|---|
| $\pi_0$ (Black et al., 2024) | 55.4 | 71.0 | 72.6 | 20.6 | 54.9 |
| $\pi_0$+FAST (Pertsch et al., 2025) | 21.2 | 74.0 | 44.8 | 15.0 | 38.75 |
| OpenVLA (Kim et al., 2024) | 68.4 | 47.4 | 59.8 | 26.6 | 50.55 |
| UniVLA (Bu et al., 2025) | 72.0 | 75.2 | 66.4 | 48.0 | 65.4 |
| OpenVLA-OFT (Kim et al., 2025) | 67.4 | 73.8 | 71.2 | 39.8 | 63.05 |
| OpenVLA-OFT + Post-training (Ours) | **85.0** | **78.4** | **78.4** | **57.8** | **74.9** |

**Effect of Termination Signals.** Table 3 further validates the effectiveness of our task success detection capability. While conventional methods rely on simulator-provided termination signals due to their inability to assess task completion, our approach employs a VLM-guided instant reflector that dynamically evaluates task success and enables early termination upon achievement. To verify this advantage, we set all compared baseline methods to strictly follow the maximum step limit for termination, whereas our framework utilizes instant reflector predictions as stopping criteria. As shown in Table 3, the compared baseline methods exhibit clear performance degradation under this setting because redundant post-success actions from delayed termination may disrupt object states after task completion (see Figure 6). In contrast, our method avoids such interference by terminating execution immediately upon detecting success signals, demonstrating our instant reflector's capacity to preserve task outcomes through timely stopping decisions.

### 5.3 LIMITATIONS AND FUTURE WORK

Despite the effectiveness of our method in enhancing VLA manipulation capabilities, it still has the following limitations. First, the performance of both world simulator and instant reflector depends on massive training data to achieve high-fidelity simulation and accurate task evaluation. Second, our VLA model optimization is slower than concurrent methods due to world simulator generation bottlenecks. In the future, we will focus on addressing these limitations.

## 6 CONCLUSION

We present a post-training framework World-Env for Vision-Language-Action (VLA) models that eliminates reliance on physical environment interaction. We introduce three core innovations: (1) RL post-training by exploration in World-Env enables policy refinement, achieving strong performance with only 5 demonstrations per task; (2) exploration in World-Env reduces physical experimentation costs; and (3) dynamic termination via VLM-guided instant reflector prevents redundant post-success actions. Experimental validation on complex manipulation tasks demonstrates our method's superiority in low-data regimes.

ETHICS STATEMENT

The adoption of world models in Vision-Language-Action systems raises practical considerations. While these models reduce reliance on real-world data collection and mitigate safety risks by enabling virtual training, they often demand substantial computational resources for training and inference. Large-scale video prediction and cross-modal alignment typically require extensive GPU usage over long durations, contributing to significant energy consumption and carbon emissions. This raises concerns about environmental sustainability, particularly when such systems are scaled or replicated across research groups without shared infrastructure or efficiency-aware design.

REPRODUCIBILITY STATEMENT

All implementation details regarding hyperparameter configurations and training protocols are detailed in Appendix B. To ensure full reproducibility of our experiments, we will make publicly available both the source code for model training and evaluation procedures, as well as pre-trained model checkpoints specifically trained on the LIBERO dataset. These resources are provided to facilitate direct replication of the experimental results presented in this work while enabling future research extensions.

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

## A  LANGUAGE MODEL USAGE STATEMENT

This paper was refined using the Qwen (Bai et al., 2023) large language model to enhance linguistic clarity, grammatical precision, and cross-disciplinary readability. No unattributed content was generated by the model; all scientific claims, data interpretations, and conclusions were independently validated by the authors.

---

**Algorithm 1** World-Env Training Algorithm

---

**Input:** Pretrained VLA policy $\pi_\theta$, scale head $\beta_\theta$, VLM-based reward function $R(\mathbf{o}_{1:t}, \mathbf{g})$, context dataset $\mathcal{D}_{\text{context}}$

1: **for** training iteration $= 1$ to $M$ **do**
2:     Set behavior policy: $\pi_\phi \leftarrow \pi_\theta, \quad \beta_\phi \leftarrow \beta_\theta$           ▷ Fix old policy and scale head
3:     Initialize rollout buffer $\mathcal{D}_{\text{rollout}} \leftarrow \emptyset$
4:     **while** $|\mathcal{D}_{\text{rollout}}| < B$ **do**            ▷ Rollout Collection
5:        Sample context $\mathbf{c} = (\mathbf{g}, \mathbf{o}_1, \mathbf{s}_1) \sim \mathcal{D}_{\text{context}}$
6:        **for** $n = 1$ to $N$ **do**          ▷ Generate $N$ rollouts per context
7:           Initialize trajectory $\tau_n \leftarrow (\mathbf{o}_1, \mathbf{s}_1)$
8:           **for** $t = 1$ to $T$ **do**
9:              Predict base action: $\boldsymbol{\mu}_t \leftarrow \pi_\phi(\mathbf{o}_{1:t}, \mathbf{s}_{1:t}, \mathbf{g})$
10:          Predict log-scale: $\boldsymbol{\beta}_t \leftarrow \beta_\phi(\mathbf{o}_{1:t}, \mathbf{s}_{1:t}, \mathbf{g})$
11:          Sample action: $\mathbf{a}_t \sim \text{Laplace}(\boldsymbol{\mu}_t, \exp(\boldsymbol{\beta}_t))$
12:          Compute next proprioceptive state: $\mathbf{s}_{t+1} \leftarrow \text{FK}(\mathbf{s}_t, \mathbf{a}_t)$
13:          Predict next observation: $\mathbf{o}_{t+1} \leftarrow \text{WorldSim}(\mathbf{o}_t, \mathbf{s}_{t+1})$
14:          Append $(\mathbf{a}_t, \mathbf{o}_{t+1}, \mathbf{s}_{t+1})$ to $\tau_n$
15:          **if** $R(\mathbf{o}_{1:t+1}, \mathbf{g}) > \eta$ **then**       ▷ Termination check ($\eta = 0.5$)
16:             $t_{\text{end}} \leftarrow t + 1$; **break**
17:          **end if**
18:          **end for**
19:          Set trajectory reward: $R_n \leftarrow R(\mathbf{o}_{1:t_{\text{end}}}, \mathbf{g})$
20:          Store log-probabilities $\log p_\phi(\mathbf{a}_{1:t_{\text{end}}} \mid \cdot)$ for importance weighting
21:        **end for**
22:        Compute RLOO baselines: $b_n \leftarrow \frac{1}{N-1} \sum_{j \neq n} R_j$ for all $n$
23:        Compute advantages: $A_n \leftarrow R_n - b_n$
24:        Add $\{(\tau_n, A_n, \log p_\phi(\cdot))\}_{n=1}^{N}$ to $\mathcal{D}_{\text{rollout}}$
25:     **end while**
26:     **for** optimization step $= 1$ to $K$ **do**
27:        Sample batch from $\mathcal{D}_{\text{rollout}}$
28:        Compute current log-probabilities $\log p_\theta(\mathbf{a} \mid \cdot)$
29:        Compute importance ratios: $r_t \leftarrow \exp(\log p_\theta - \log p_\phi)$
30:        Update $\pi_\theta$ and $\beta_\theta$ by minimizing PPO loss (Eq. 5)
31:     **end for**
32: **end for**

---

## B  MORE IMPLEMENTATION DETAILS

### B.1  DEATILS OF SCALE HEAD

Our method builds upon OpenVLA-OFT (Kim et al., 2025), which predicts continuous actions via an action head that takes hidden states $f \in \mathbb{R}^d$ as input and employs L1 loss for action regression:

$$\mathcal{L}_{\text{L1}} = \|\mathbf{a}_{\text{gt}} - \boldsymbol{\mu}\|_1 \quad \text{where } \boldsymbol{\mu} = \text{MLP}_{\text{action}}(f). \tag{6}$$

To model heteroscedastic uncertainty in action prediction, we introduce a scale head with the same MLP architecture as the action head, as shown in Figure 7. This scale head outputs log-scale parameters $\boldsymbol{\beta}$ through:

$$\boldsymbol{\beta} = \text{MLP}_{\text{scale}}(h), \tag{7}$$

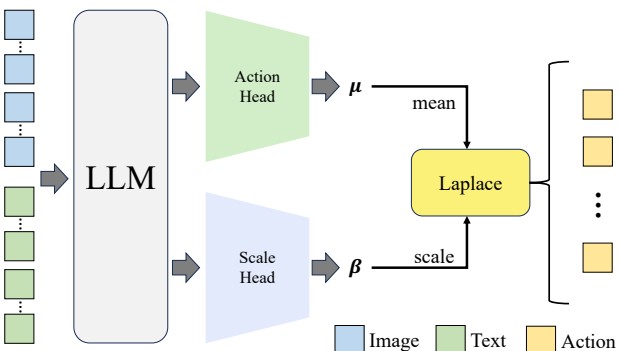

Figure 7: **Architecture for uncertainty-aware action generation.** The deterministic action output of the VLA policy is augmented with a parallel Laplace scale head to model action uncertainty.

and is trained with negative log-likelihood (NLL) loss under a Laplace distribution assumption:

$$\mathcal{L}_{\text{NLL}} = \underbrace{|\mathbf{a}_{\text{gt}} - \boldsymbol{\mu}| \cdot e^{-\boldsymbol{\beta}}}_{\text{Data fit}} + \underbrace{\boldsymbol{\beta}}_{\text{Uncertainty penalty}} + \log 2. \tag{8}$$

The scale head is trained using a batch size of 8 and a learning rate of $5 \times 10^{-4}$ over $1,000$ training iterations.

### B.2 DEATILS OF WORLD SIMULATOR

We adopt the original implementation of the EVAC world model (Jiang et al., 2025b) and retain its training configuration. We show an overview in Figure 8. The generation process starts from a reference image, whose CLIP (Radford et al., 2021) features provide style guidance. This signal is integrated into the diffusion model via cross-attention. Action information is encoded as a spatial action map and concatenated with visual features at the feature level. The fused representation drives the diffusion network to generate future frames through iterative denoising, followed by a video decoder to produce the final output. The EVAC model was originally designed for dual-arm robotic platforms with 14-dimensional (14D) action vectors (7D per arm). In contrast, the LIBERO benchmark employs a single-arm robot with 7D actions (6D end-effector pose + 1D gripper state). To maintain compatibility with the EVAC architecture, we zero-pad the unused 7D action dimensions during training, preserving the input interface while adapting to the target hardware.

### B.3 DETAILS OF REWARD HEAD

Our VLM-guided instant reflector integrates a pretrained vision-language model (LLaVA (Liu et al., 2023c)) with a lightweight reward head that predicts continuous reward signals, see Figure 9 for an overview. The VLM backbone is kept frozen to preserve its semantic capabilities, and only the reward head is trained. Given a video sequence $\{f_1, \ldots, f_N\}$ generated by the world simulator, we uniformly sample 32 frames as visual input. The language prompt is formatted as: *"Watch the video and determine whether it completes the task: $\{\mathbf{g}\}$ — answer only 'Yes' or 'No'."* The VLM processes this input and extracts a pooled embedding, which is projected by the reward head to a scalar. A sigmoid activation yields a continuous reward $R \in [0, 1]$, interpreted as the task completion probability. The reward head is trained with binary cross-entropy loss, using a batch size of 8, learning rate $1 \times 10^{-4}$, Adam optimizer, and 50 epochs, with input frames center-cropped to $384 \times 384$ resolution.

## C ANALYSIS OF WORLD SIMULATOR

### C.1 DATA ANALYSIS AND DISTRIBUTION

We provide a statistical analysis of the training data for the world simulator and instant reflector in Figure 10, including: (a) length distributions for successful vs. failed trajectories, (b) cumulative

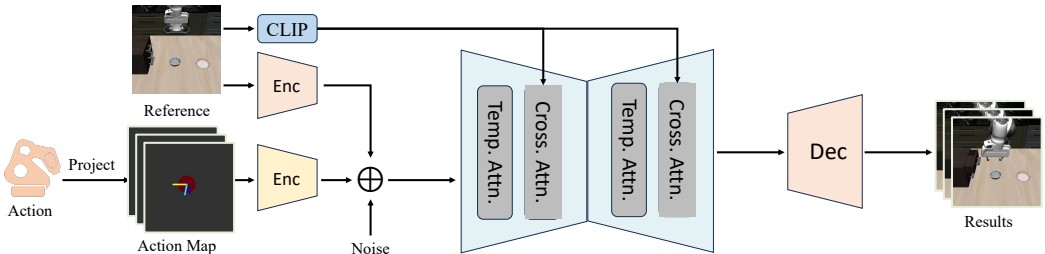

Figure 8: **Overview of the world simulator.**

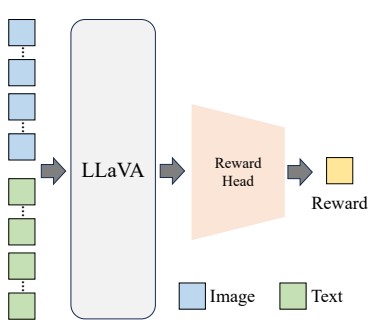

Figure 9: **Network architecture of instance reflector.**

distribution functions by outcome, and (c) task outcome proportions. The bimodal distribution in successful trajectories motivated our dynamic termination mechanism, while the long-tailed length distribution informed our curriculum sampling strategy.

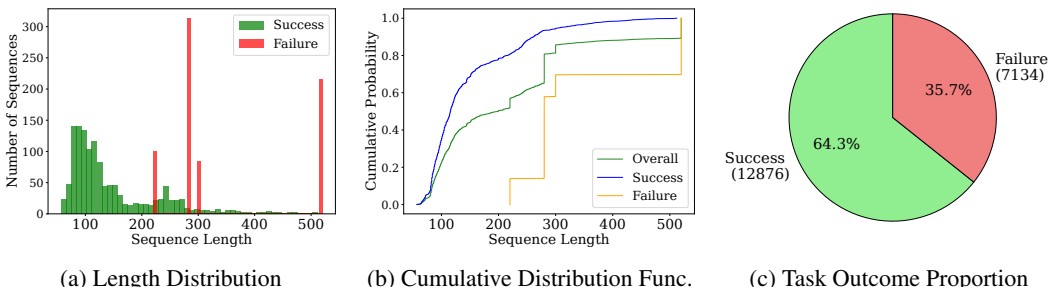

(a) Length Distribution     (b) Cumulative Distribution Func.     (c) Task Outcome Proportion

Figure 10: **Training data analysis and distribution.**

### C.2 MORE RESULTS OF WORLD SIMULATOR

Figures 11 and 12 show additional trajectories generated by the world simulator, demonstrating its ability to synthesize both successful and failed task executions.

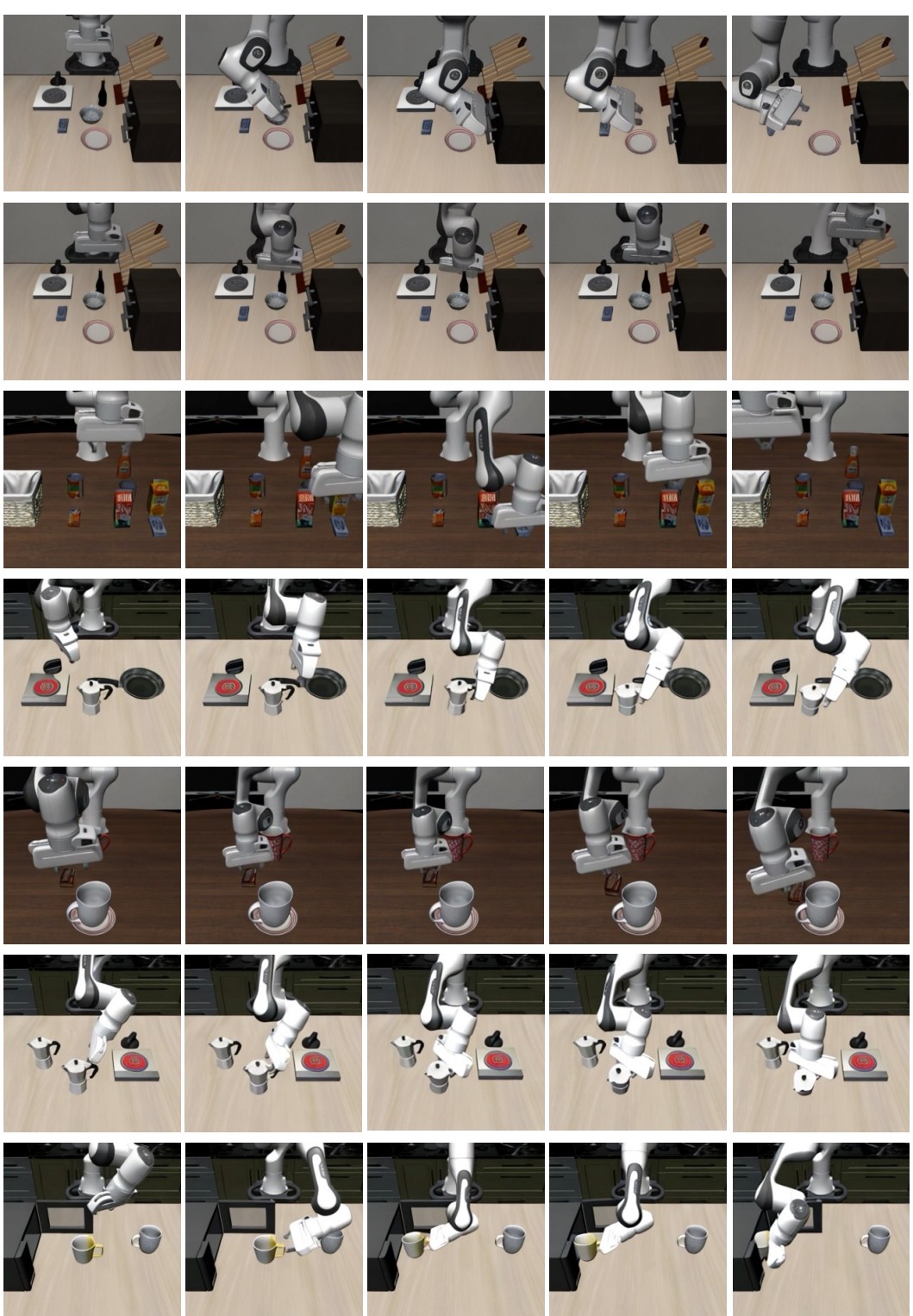

Figure 11: **Failure trajectories synthesized by the world simulator.**

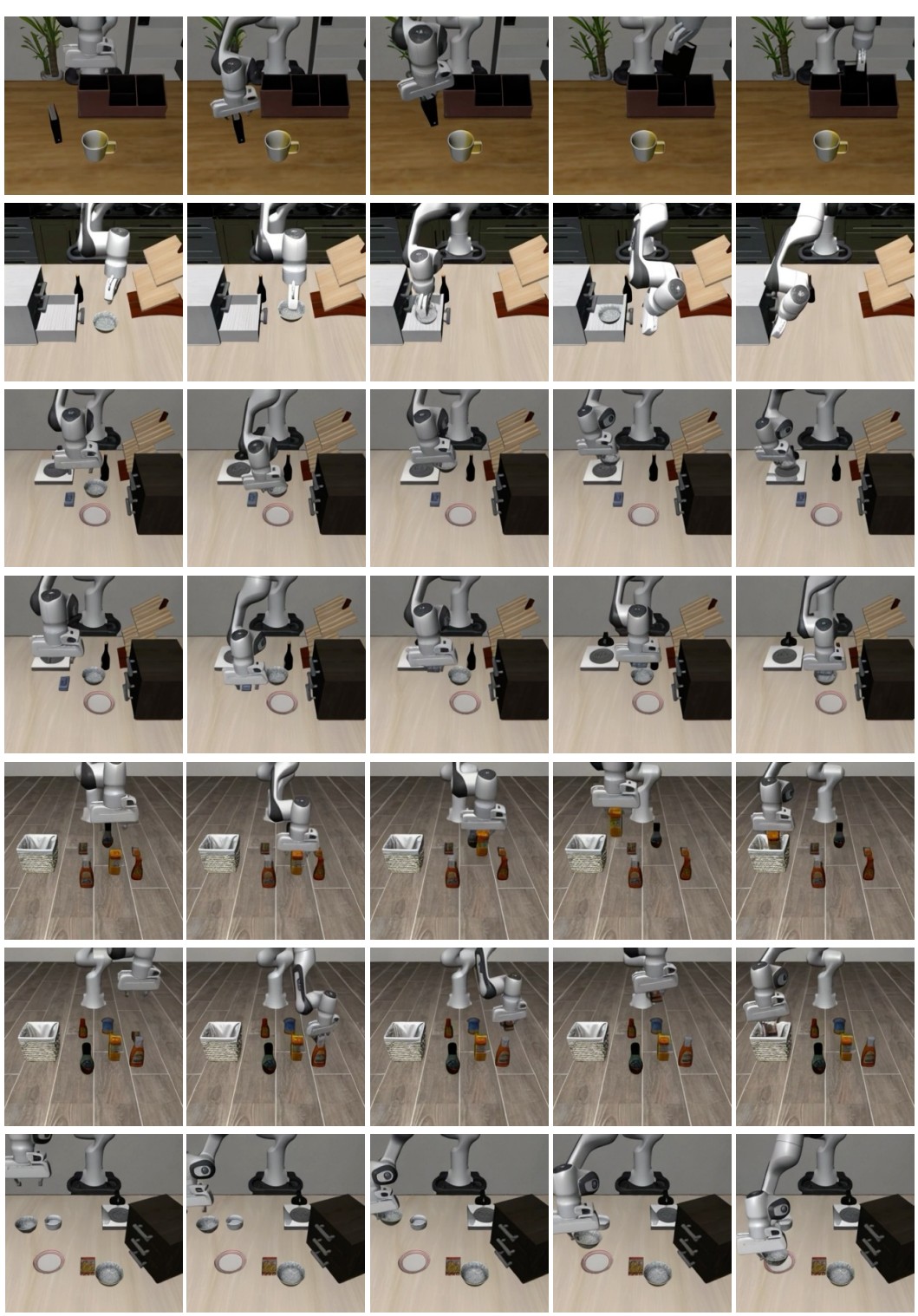

Figure 12: **Success trajectories synthesized by the world simulator.**

