# OpenReview forum: "World-Env: Leveraging World Model as a Virtual Environment for VLA Post-Training"
_ICLR.cc/2026/Conference — ICLR 2026 Conference Withdrawn Submission_

### Official Review · Reviewer_M2Kt · 2025-10-26

**Soundness:** 3
**Presentation:** 3
**Contribution:** 2
**Rating:** 4
**Confidence:** 5

**Summary:**

This paper presents World-Env, a reinforcement learning (RL) framework that leverages a video-based world model as a low-cost virtual simulator for Vision-Language-Action (VLA) post-training. The system integrates two modules: a world-model-based action-conditional video generator and a Vision-Language Model (VLM)-guided instant reflector, which provides continuous rewards and early termination signals.
By replacing physical interaction with a world model, the framework allows safe, data-efficient policy optimization and enables the formation of a closed-loop RL training pipeline. Experiments on the LIBERO benchmark show that World-Env achieves competitive performance with as few as five demonstrations per task, demonstrating improved sample efficiency and training stability.

**Strengths:**

1. The paper explores an important direction for the VLA community: how to construct a closed RL loop through action-conditioned video generation to simulate policies and use VLM-based evaluators to define rewards. This contributes to the broader goal of realizing self-improving embodied agents without expensive real-world rollouts.

2. The proposed RL loop demonstrates tangible benefits in few-shot settings. On LIBERO, the method achieves noticeable gains over strong baselines using only limited demonstrations, confirming the practical effectiveness of the world-model-based post-training framework.

3. The paper is clearly structured, easy to follow, and technically transparent. The training procedure, data strategy, and optimization loop are well described (Figures 2–4), making the approach easy to reproduce and extend.

**Weaknesses:**

1. While the system design is interesting, its core components are adaptations of existing architectures. The world simulator builds upon the EVAC action-conditional video generator, and the instant reflector is an incremental modification of an existing VLM (LLaVA) with a lightweight reward head. The contribution therefore, lies primarily in system integration rather than fundamental algorithmic innovation.

2. The claimed advantage of replacing the simulator with a world model is not convincingly demonstrated. All experiments are conducted within a simulator environment (LIBERO), which already provides accurate physical rollouts. A direct comparison with “Simulator-RL” (i.e., RL conducted using simulator-generated observations and rewards) is missing and essential for validating whether the world-model-based simulation truly offers an advantage beyond data efficiency.

3. The paper shows real-scene renderings (Figure 5) but lacks quantitative real-world metrics or a demonstration video. The manuscript would benefit from actual robotic experiments that close the loop between simulated policy optimization and real execution.

4. Although the authors claim reduced cost via virtual rollouts, the paper omits runtime and compute statistics (e.g., frame generation latency, total GPU hours per iteration). Without this, the practical efficiency relative to standard simulator-based RL remains unclear.

5. The instant reflector’s termination threshold (η = 0.5) is fixed and not justified. It would be useful to see sensitivity analyses or calibration experiments to ensure stability across tasks and instructions.

**Questions:**

1. Can the authors quantify the generation overhead per step and compare the total training time with the standard simulator-based PPO?

2. How was the termination threshold (η = 0.5) chosen? Does performance vary significantly across different threshold values?

---

### Official Review · Reviewer_KSSx · 2025-10-30

**Soundness:** 1
**Presentation:** 3
**Contribution:** 2
**Rating:** 2
**Confidence:** 4

**Summary:**

This paper presents a method for post-training a VLA-style model using a video-world model. They replace real-world rollouts during RL post-training with rollouts from their video-based world model. It also proposes using a VLM-based “instant reflector” for reward and termination estimation inside their world model.

**Strengths:**

The methods section and presentation quality are strong. The paper identifies an important problem: the high cost of rollouts in real-world RL.

**Weaknesses:**

I am assigning my score because the work argues that real-world rollouts are prohibitively costly, but the approach itself depends on them to train a world model, and the extent of this dependence is not clear to me.  More specifically:
* The paper claims that post-training in real settings is costly due to difficult resets and the cost of failures. However, the world model is trained using exploration data collected by rolling out the policy with injected noise. This appears to require the same kind of rollouts that the paper claims are infeasible. Comparing the ablation results training without policy rollouts shown in Table 2, it performs worse than the OFT base policy.
* The paper does not clearly report the number or cost of the on-policy exploration rollouts. In Appendix Fig. 10, it shows that ~20k trajectories (including ~7k failures) are used to train the world model, but the breakdown across tasks and between expert vs. policy-generated data is not provided.
* The evaluation does not include comparisons to RL-based post-training methods such as RIPT-VLA [2]. Such a comparison, ideally under the same rollout budget, would help clarify the practical tradeoffs.
* Training a reward classifier is common in past real-world RL works [1] when there is no oracle termination signal. It seems more of a necessity because of a lack of success signals from the world model. This work uses oracle termination signals from the environment to train its instant reflector, meaning WorldEnv requires access to oracle success/failure signals.
* The final reported performance numbers in Tables 1–3 do not include standard deviations, confidence intervals, or the number of seeds.
* Equation (2) has a possible typo, as it does not match the deterministic policy gradient. It seems to be misapplying the chain rule. If it is non-deterministic, it is missing a log


[1] Jianlan Luo, Charles Xu, Jeffrey Wu, Sergey Levine. Precise and Dexterous Robotic Manipulation via Human-in-the-Loop Reinforcement Learning

[2] Shuhan Tan, Kairan Dou, Yue Zhao, et al. RIPT-VLA: Interactive Post-Training for Vision-Language-Action Models.

**Questions:**

* Does the choice of a VLM backbone, rather than a pretrained ResNet trained per task, for the termination module make a difference? How much does this learned termination impact performance compared to the oracle termination signals?
* How many world-model training rollouts were collected in total? What kinds of resets and failures did these experience?
* How large was the effect of the stochastic exploration coefficient?
* Can you report performance with oracle termination signals to quantify how much of an effect the termination module has?
* Do you start from the fine-tuned model or the base model when performing RL in your world model?
* The increase from fine-tuning with just 5 expert demonstrations seems larger than that from post-training with World-Env on top of the fine-tuned base model. How many more expert demonstrations does ft require to match the world-env post-training, and what does the improvement look like on top of a base VLA with no fine-tuning? (if this work performs RL on top of the base policy, then disregard this question)
* What happens if you take the successful rollouts used in the world model training and perform supervised fine-tuning with them?

---

### Official Review · Reviewer_PjvM · 2025-11-01

**Soundness:** 2
**Presentation:** 3
**Contribution:** 2
**Rating:** 4
**Confidence:** 4

**Summary:**

The authors proposed World-Env, (1) a video-based world simulator (built on EVAC) that predicts next visual observation conditioned on action and proprioception; (2) a frozen VLM (LLaVA) plus a lightweight reward head that outputs a scalar in [0,1] and triggers termination when above a threshold. Policy is OpenVLA-OFT with an uncertainty-aware Laplace scale head; RL uses LOOP (RLOO + PPO) with continuous rewards.

**Strengths:**

1. Strong motivation and practical value in non-resettable, safety-critical settings.
2. (Almost) Clear, reproducible system with algorithmic details and hyperparameters.
3. Data augmentation through self-exploration improves world-model robustness; supported by ablations.

**Weaknesses:**

1. Innovation is largely in system integration;
2. Lack of rigorous physical-consistency evaluation for the world model; risk of “pretty but wrong” dynamics.
3. no matched-budget RL baselines and limited gains on long-horizon tasks.
4. The real-world demos are quite limited and focusing on the table-top manipulation tasks. Figure 5 shows the generation results, not the manipulation task execution tasks. Also, the comparison with other models in the real-world environment is missing.

**Questions:**

1. The EVAC produces only the visual frames, but the action policy (Line 155) needs also the joint states, how do you get these?

2. Line 291, from the same initial states and with the same policy model, how is the generated rollouts similarity? If they are too similar, can the PPO find the advantages inside a single batch?

3. For the real-world experiments, can you give more details on the data collection process (using tele-operation?) and the final dataset statistics?

4. How is the proposed methods applied in the real-world VLA model?

5. The author claimed that the usage of the simulator's binary signal would bring the harm that "the lack of termination-aware feedback, causing policies to often continue executing redundant actions after task completion", so they chose to use a VLM instead. However, the training samples seems also come from the simulator? Then would the obtained VLM bias to that?  Further discussions is needed.

---

### Official Review · Reviewer_1cuT · 2025-11-02

**Soundness:** 2
**Presentation:** 2
**Contribution:** 2
**Rating:** 4
**Confidence:** 2

**Summary:**

The paper proposes World-Env, a post-training framework to finetune pretrained VLA models on a learned world model and reward / done model (reflector).

**Strengths:**

+ The paper is well-written and presented

**Weaknesses:**

- While table 2 shows an improvement over the selected baselines, none of the baselines have an equivalent RL finetuning stage as the presented approach, making the comparison unfair.
- The paper lacks experiments showing its efficacy on real-world setting. While its compelling to see the model works on synthetic environments, one can directly use the simulator for finetuning after all.
- There is a lack of performance analysis on the learned world / reward model. For example, how long can the learned world model rollout forward before its prediction starts to get unrealistic? What are the accuracy / F1-score of the learned reward model, given different rollout horizon and training budget? And how do all of these metrics relate to an improvement over the final VLA's performance?

**Questions:**

Similar to the weakness section above. Additionally, can the authors comment on
1. How well would the world model and the entire presented approach work if the observation also includes ego-motion? E.g. the camera pose moves from ego-motion.
2. What is the asymptotic performance of the approach? If not reaching 100% success rate, what might be the bottlenecks / how to tackle them?

---

### Note · Authors · 2025-11-12

I have read and agree with the venue's withdrawal policy on behalf of myself and my co-authors.